# Preparation and Performance Characterization of an Active Luminous Coating for Asphalt Pavement Marking

**Kaifeng Wang** [1,2], **Ziyu Lu** [1,2], **Yingxue Zou** [3], **Yunsheng Zhu** [1,2,*] and **Junhui Yu** [1,4]

1 School of Transportation and Logistics Engineering, Wuhan University of Technology, Wuhan 430063, China
2 Hubei Highway Engineering Research Center, Wuhan 430063, China
3 State Key Laboratory of Silicate Materials for Architectures, Wuhan University of Technology, Wuhan 430070, China
4 China Railway Fourth Survey and Design Institute, Suzhou 215000, China
* Correspondence: zhuyunshengwhlgdx@163.com

**Abstract:** For improving the night recognition of road markings and enhancing the driving safety of asphalt pavements, single-factor optimization is used to investigate the effects of the component materials, including luminescent power, pigment, filler, and anti-sedimentation agent, on the luminous performance of a coating. Additionally, their composition ratios are optimized using response surface methodology. A phosphorescent marking coating is prepared to investigate the micromorphology, excitation, and emission properties using scanning electron microscopy (SEM) and molecular fluorescence spectroscopy (MFS). The optimum thickness of the coating on an asphalt pavement is investigated, and the durability of the coating on asphalt pavement using a wheel rutting test is evaluated. The results show that the 300 mesh yellow-green luminous powder has the optimal overall performance, with an initial luminescence that exceeds that of orange and sky blue by three times. Initial brightness is mainly influenced by aluminate luminescent powder (ALP), which increases with the dosage. ALP and fumed silica powder (FSP) have a positive effect on brightness after centrifugation, and the effect of FSP dosage is more significant. ALP, rutile titanium dioxide powder (RTDP), and FSP influence the wear value of the coating, and the magnitude of the effect is RTDP > FSP > ALP. The optimal dosages of the main component are 27% ALP, 5% RTDP, and 0.8% FSP. The results of SEM show that the components in the coating are evenly dispersed, and the surface of the coating is rough. The peak excitation wavelength of 420 nm means that the coating has the best excitation effect in UV light, and its emission spectrum in the 440–760 nm wavelength range is well within the sensitive recognition zone of the human eye. The initial brightness gradually reached 4.38 cd/m$^2$ when the coating thickness was increased from 482 μm to 546 μm, and the optimal application thickness of the luminous coating was determined to be 500 μm. At high and normal temperatures, the rutting stripping rates of the luminous marking coating are 16.8% and 8.2%, indicating its satisfactory durability. This study provides an experimental basis for the ratio optimization design of a luminous coating for asphalt pavements.

**Keywords:** luminous coating; road marking; luminescent property; phosphorescent; response surface methodology; asphalt pavement

## 1. Introduction

As an integral part of the road traffic system, road marking is a coating system consisting of color areas and reflective areas which are considered the main tool for delineating and directing traffic, and warning, managing, and informing drivers [1]. It provides traffic participants with a clear preview of the outline of the traffic lane during the day, through the high visibility of the white pigment. In the absence of road lighting, glass beads work by preferentially reflecting the light from vehicle headlights in the opposite direction, performing the function of indicating road traffic [2,3]. Nevertheless, the commonly used reflective

road markings cannot fully meet the traffic safety needs in a dark environment, and the passive road guidance effect is greatly influenced by the surrounding environment and the light from the vehicle's headlights. Glass beads' effective light reflectivity decreases when the light is diminished in rain and fog because the water film covers them and creates a diffuse reflection phenomenon. The number and severity of crashes are significantly higher than during the day because of the lower quantity and quality of available visuals [4]. The low visibility of the night environment and the limited guiding ability of road markings are the main risk factors triggering traffic accidents at night [5,6]. In the United States, the annual number of fatalities has risen by 11% compared to 2010, reaching 38,800 people. This has resulted in a huge social cost, with economic losses estimated to even reach 3% of the world's GDP [7]. The risk of fatalities from traffic crashes at night is 3–4 times higher than that in the day, and 67% of all traffic crashes occur between 6 and 10 p.m. [8,9]. The improvement and maintenance of traffic signs can reduce the incidence of night traffic accidents by 50% and reduce the number of fatalities in nighttime accidents by about 28% [10]. Consequently, research into a road marking that may actively light up at night is crucial from a practical standpoint in order to decrease the frequency of nighttime traffic accidents.

A solution proposed for this purpose is to add phosphorescent materials to the road markings [11]. When a phosphorescent material is excited by a light source, energy is stored in the form of captured excited state electrons or holes and is slowly released at night to achieve self-luminescence. Sustained photoluminescence is considered a photo-responsive phenomenon that can be exploited to give objects the ability to emit light in the dark [12]. Relevant engineering practices at home and abroad also show that the use of phosphorescent markings can play an obvious warning role in accident-prone road sections such as ramps, curves, tunnel entrances, and exits [13]. The sulfide series, silicate series, and aluminate series are the most typical phosphorescent materials. Compared with other series, aluminate series' phosphorescent materials are the most commonly used phosphorescent materials, and the preparation process is relatively mature [14]. Used as one of the most widely used light-emitting materials, ZnS: Cu, as the representative of sulfide phosphorescent materials, has been developed significantly. However, later, it was gradually eliminated because of the poor performance of the remaining glow and it also having certain radioactivity [15]. Subsequently, researchers found that the long afterglow phosphorescent material $SrAl_2O_4$: $Eu^{2+}$ already had a good luminous performance, but the afterglow performance was not ideal [16]. Therefore, the element "Dy" was introduced in the study of long-lasting phosphorescent materials. As one of the brightest known long-lasting phosphorescent materials, the afterglow intensity of strontium aluminate ($SrAl_2O_4$: $Eu^{2+}$, $Dy^{3+}$) doped with europium and dysprosium reaches ten times that of ZnS: Cu, Co phosphors. After two weeks of continuous UV irradiation at 370 nm, the luminous intensity can still be preserved by 80%, with strong resistance to photobleaching [17]. The luminous principle of this material is as follows: Since the ionic radius of $Dy^{3+}$ is larger than that of $Sr^{2+}$, when a part of $Sr^{2+}$ is replaced, the lattice shape of the product changes, and a trap level is created. Free electrons are produced and kept in the trap when energized by the light source. The trapped trap electrons are released once photoexcitation stops, and they combine with the luminous center, utilizing thermal interference to create an after-glow at an ambient temperature [14,18]. Different rare earth ions have different atomic numbers, and the location and effectiveness of the defect energy levels after replacing $Sr^{3+}$ are different. Therefore, under the same excitation conditions, the brightness and time of the afterglow cannot be the same [19].

In addition, due to the shortcomings of heavy metal leaching and VOCs volatilization, traditional solvent-based coatings are gradually replaced by environmentally friendly water-based coatings [20,21]. In addition, waterborne road marking coatings have a higher fire risk, good dust resistance, and better hygiene during production and application [22]. Nance et al. compared the durability, transparency, adhesion, compatibility, and dispersion of various waterproof coatings with $SrAl_2O_4$: $Eu^{2+}$, $Dy^{3+}$. A good coating will be compatible with the substrate and carrier, and the coating system must adhere well to the phosphor

to protect it from hydrolysis [23]. Bi et al. prepared an acrylic copolymer emulsion using a homemade modified luminescent powder, light $CaCO_3$, talc powder, dispersing agent, anti-foaming agent, anti-sediment agent, and film-forming additive, and optimized the design of its formulation. They found that the coating with sodium hexametaphosphate and sodium carboxymethylcellulose as additives exhibited a satisfactory performance when mixed with silicone emulsions [24]. Studio Roosegaarde in the Netherlands introduced the concept of luminous pavements and used it as a theory to build a test pavement, the Oss N329 luminous pavement [25]. Previous research on the luminous properties of the luminous material itself has made great progress, but the single-factor and multi-factor coupling influence of ingredients in the coating is still ambiguous [26]. Additionally, the compatibility of luminous material with pigments, fillers, anti-sedimentation agent, anti-foaming agent and leveling agent is less studied. The existing research is mainly focused on the application of decorative and safety signs and less on the application of road marking coatings [27]. Filling this research gap is of great significance to improve the durability and prolong the service life of the coating.

Therefore, the effects of the component materials within the coating on the luminescence, afterglow, wear resistance, and adhesion performances of the coating at different dosages were investigated using single-factor optimization, including luminescent powder, film-forming materials, pigments, fillers, film-forming auxiliaries, anti-sedimentation agents, anti-foaming agents, and leveling agents. The methodological framework is illustrated in Figure 1. After determining the important influencing factors, the experimental design was carried out using response surface methodology and the best ratio was analyzed by establishing a fitted model. Based on this ratio, a phosphorescent marking coating was prepared to establish the basic physical properties, micromorphology, excitation, and emission properties of the coating. The effect of different coating thicknesses on performance was studied to determine the optimum thickness of the coating on asphalt pavements, and its durability was analyzed.

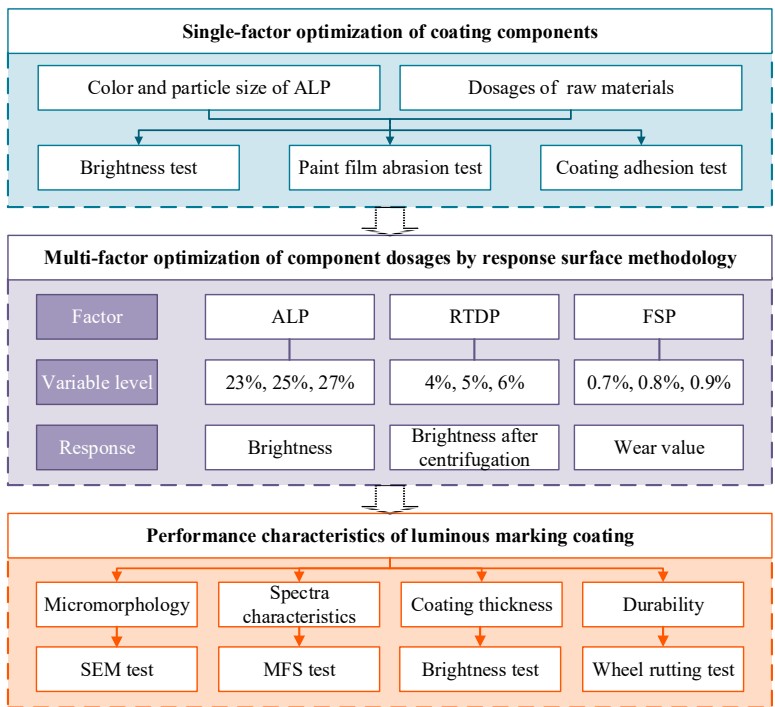

**Figure 1.** Methodological framework.

## 2. Materials and Experimental Methods

### 2.1. Raw Materials

2.1.1. Phosphorescent Material

Aluminate luminescent powder (ALP) produced by Foshan Juliang Optical Material Co., Ltd. (Foshan, Guang Dong, China) was applied as the phosphorescent material, and its main properties are shown in Table 1. At present, the commonly used luminous powder colors are mainly yellow-green and sky blue [28,29]. Three kinds of ALP with yellow-green, sky blue, and orange colors were selected for coating preparation and experimental comparison. Four kinds of yellow-green ALP (100 mesh, 200 mesh, 300 mesh, and 400 mesh) were selected to prepare the coatings for relevant performance testing.

**Table 1.** Properties of luminescent powder.

| Properties | Unit | Results | Methods [30–32] |
|---|---|---|---|
| Particle size | mesh | 100–400 | / |
| Density (25 °C) | g·cm$^{-3}$ | 3.2 | GB/T 6750-2007 |
| Stimulation time | min | 10–15 | GB/T 24981.2-2020 |
| Luminous time | h | 8–10 | GB/T 24981.2-2020 |
| Decomposition temperature | °C | 800 | GB/T 9269-2009 |

2.1.2. Film-Forming Materials

Silicone-modified styrene-acrylic emulsion (SSAE) with a solid content of 48.2% produced by Qingdao Ocean Haidun Co., Ltd. (Qingdao, Shan Dong, China) was employed as the film-forming material. To extend the ambient temperature applicability of water-based coatings in engineering applications, alcohol ester-12 (AE-12) was added at 3% dosage as a film-forming auxiliary.

2.1.3. Pigments and Fillers

Rutile titanium dioxide powder (RTDP) with a stable chemical property was selected as the pigment of the luminous road marking paint [33,34]. Ultrafine glass powder (UGP) was used as filler because it has excellent light transmission and reflection properties, which helps to give full play to the light absorption, light storage, and luminescence properties of photoluminescent coatings [35].

2.1.4. Auxiliaries

Fumed silica powder (FSP) was chosen as the anti-sedimentation agent in this study because it is more suited for acrylic resins and has no negative effects on luminescent powder [36]. Mineral oil polyether (MOP) was used to obtain an excellent defoaming performance as the anti-foaming agent, and fluorine-modified polyacrylate (FMP) was used as the leveling agent. The dosages of these auxiliaries were 0.6%.

### 2.2. Experimental Methods

2.2.1. Coating Treatment of Luminescent Powder

Aiming to improve the water-resistance of alkaline earth aluminate luminescent powder and maintain the acid-base balance of the system, the physical coating method of silica by dissolving sodium silicate in a small amount of hot water to form a strongly alkaline solution to form $Si(OH)_4$ colloid was adopted. After the addition of acid, $Si(OH)_4$ colloidal particles grow and precipitate out. The generated $Si(OH)_4$ was deposited and formed a bonding point with the surface of aluminate particles, which is shown in Figure 2. Then, it continued to attract $Si(OH)_4$ on the bonding point and condensed into silica hydrate. It gradually formed a dense silica cladding layer, and finally encapsulated the aluminate closely [37]. In order to obtain a uniformly dispersed and stable luminous road marking coating, it is necessary to ensure that the solid components within the coating are able to mix well with the liquid components and that no solid precipitation occurs.

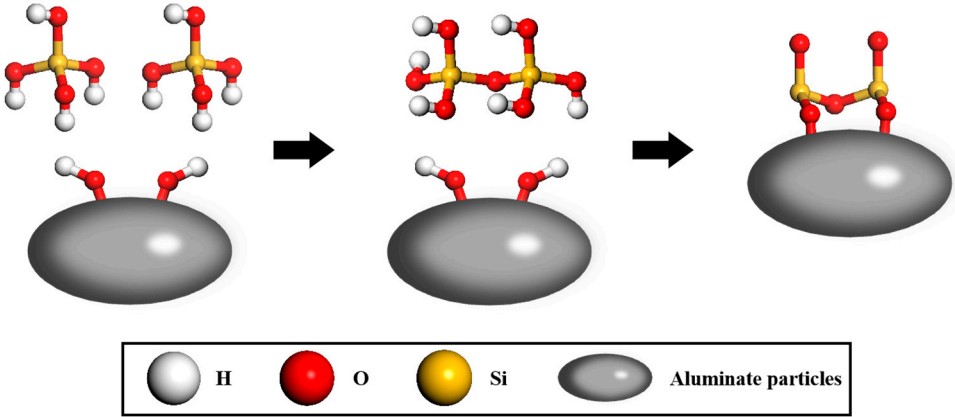

**Figure 2.** Coating process of silicon hydroxide colloid on the surface of the luminescent powder.

### 2.2.2. Preparation of Phosphorescent Paint

First of all, a quantitative dosage of SSAE was weighed, and half of FSP was added and mixed to form the emulsion mixture. After that, the other half of the FSP was mixed with ALP, RTDP, and UGP to form a filler mixture. In addition, the selected AE-12, MOP, and FMP were divided into two parts with the filler mixture and added into the emulsion mixture in two batches to facilitate the dispersion of solid components [25]. In the last stage, the coating pH was adjusted to 8–9 using an alkaline solution [38].

### 2.2.3. Characterization of the Luminous Marking Coating

For quantitative evaluation of the effect of components in the phosphorescent coating on the coating performance, the mass of SSAE was chosen as the quantification in the preparation of the coating. The dosage of the remaining components was defined as the ratio of their weight to the weight of SSAE.

The brightness test, paint film abrasion test, and coating adhesion test were employed to investigate the effect of each component on the performance of the luminous marking coating. The luminous test is aimed at quantitatively characterizing the initial luminosity and the afterglow performance of the luminescent material over time [32]. First, the test coating was applied to a glass disc and left to form a film. It was placed in the dark at room temperature for 24 h to fully release the light, and then irradiated under a standard light source D65 for 10 min [39]. After removing the light source, the initial luminosity of the sample was immediately tested with a CS-200 spectrophotometer. The afterglow luminosity was tested at 10 min, 20 min, 30 min, 40 min, 50 min, and 60 min, respectively.

The prepared paint was evenly coated on a glass disc with a diameter of 10 cm, a thickness of 3 mm, and a middle gouge diameter of 7 mm. The sample was maintained in an intelligent constant temperature and humidity incubator with temperature maintained at $(23 \pm 2)$ °C and humidity maintained at $(50 \pm 5)$ % for 24 h. The maintained specimen was fixed on the turntable of the paint film abrasion instrument, and a 1000 g weight was attached to the pressurized arm. The number of revolutions was adjusted (50 revolutions for the first time and 200 revolutions for the second time) and the main switch, the vacuum cleaner switch, and the turntable switch turned on in turn for sanding [40]. The masses of the samples after the first and second abrasions were weighed separately and recorded as $m_1$ and $m_2$, and the abrasion value was $m_1 - m_2$ [41].

The adhesion of the coatings was evaluated indirectly using the scribing method [42]. The prepared paint was applied on a tinplate of 120 mm in length and 50 mm in width. It was maintained in an intelligent constant temperature and humidity incubator at $(23 \pm 2)$ °C and $(50 \pm 5)$% for 24 h. The maintained specimen was fixed on the paint film adhesion tester, pressed with 200 g weight, and the handle was turned clockwise at a constant speed (70–100 rpm); the upper side of the scratch on the sample was the target of inspection. The seven parts from 1 to 7 were marked in turn and accordingly divided into seven grades, with grade 1 being the best and grade 7 being the worst. If more than 70% of

the grid of a part was intact, the part was considered to be qualified and was evaluated as the grade corresponding to the current part [43]. Five parallel tests were performed on all test samples and the results were displayed with error bars.

### 2.2.4. Experimental Design with Response Surface Methodology (RSM)

Adopting the Box–Behnken design method for response surface methodology, a three-factor, three-level, three-response experiment was conducted [44]. Following the experimental results of the single-factor optimization design, ALP, RTDP, and FSP were used as the three main factors affecting brightness, brightness after 15 min centrifugation, and wear value. Three center points were designed and a quadratic model was built using non-linear regression fitting, as shown in Equation (1). It was used to link all the process variables, to explain the experimental results, and to produce response surface plots.

$$Z_n = a_0 + \sum_{i=1}^{k} a_i X_i + \sum_{i=1}^{k} a_{ii} X_i^2 + \sum_{1 \leq i \leq j}^{k} a_{ij} X_i X_j + \varepsilon \tag{1}$$

where $Z_n$ = response values ($n$ = 1,2,3), $a$ = regression coefficient, $X$ = input factor, and $\varepsilon$ = error. $Z_1$, $Z_2$ and $Z_3$ refer to the response values of brightness, brightness after 15 min centrifugation, and wear value, respectively. $a_0$, $a_i$, $a_{ii}$ and $a_{ij}$ are the regression coefficients of constant, linear effect, quadratic effect, and interaction, respectively.

### 2.2.5. Usage Performance of Luminous Marking Coating

MFS was employed to obtain emission spectra and excitation spectra of luminous marking coatings to analyze whether their luminous properties met the safety requirements [45]. SEM (JSM-5610LV, JEOL) was applied to study the surface morphology of the coatings. The test conditions were as follows: the accelerating voltage was 20 kV, and the magnification was 1000 μm and 50 μm, respectively. The overall morphological characteristics of the coatings were analyzed, and the distribution of each component in the coating system was also evaluated to determine the homogeneity of the coatings. For determining the durability of the coating at high temperatures, the prepared coating was evenly coated on the rutting plate according to 15 cm width, and after 24 h of maintenance at room temperature. The rutting tester was used for 60 min of high-temperature (60 °C) rolling and normal temperature (23 °C) rolling. At last, the stripping rate of the coating on the rutted plate after crushing was quantified using image processing techniques. The calculation of the stripping rate was given in Equation (2).

$$R_{Stripping} = \frac{A_1}{A_0} \tag{2}$$

where $R_{Stripping}$ refers to the stripping rate of coating, $A_1$ refers to the exposed area of asphalt mixture, and $A_0$ refers to the total area of coating coated.

## 3. Results and Discussions

### 3.1. Single-Factor Optimization of Components in Marking Coatings

The performance of coatings prepared from luminescent powders of different colors and particle sizes was investigated. Optimal dosages of four ingredients, including ALP, pigment, filler, and anti-sedimentation agent, were determined and their effects on coating performance were evaluated with a single-factor optimization design.

### 3.1.1. Color and Particle Size of ALP

The morphology of the three colors' coatings are shown in Table 2. Yellow-green ALP is light yellow under natural light and green under emitting status. Since the orange ALP used in this study is made of yellow-green ALP and red color powder, it is red under natural light. Figure 3 illustrates the effect of different ALP colors on coating performance. The initial brightness of the luminous coating prepared from yellow-green ALP was the

largest, followed by orange ALP. In comparison, the brightness of the coating prepared with sky blue ALP was the lowest, only 26.35% of the initial brightness of yellow-green. The afterglow performance was characterized by brightness after 10 min, and the yellow-green ALP was also the best. For orange ALP, the addition of red color powder with the strong masking effect reduced the initial brightness of ALP and affected its luminous performance to a greater extent. Due to the material of the blue luminous coating being silicate, its luminous intensity was not enough for afterglow performance and was poor in its material properties. A lower wear value means better wear resistance [46]. It can be seen that the yellow-green sample had the best wear resistance, while sky-blue samples had the worst. In summary, different colors of ALP had a greater impact on the luminous performance of the luminous coating, and the yellow-green ALP was selected as the best.

**Table 2.** The morphology of different colors of luminous coatings before and after exciting.

| | Yellow-Green | Orange | Sky Blue |
|---|---|---|---|
| Natural | | | |
| Luminous | | | |

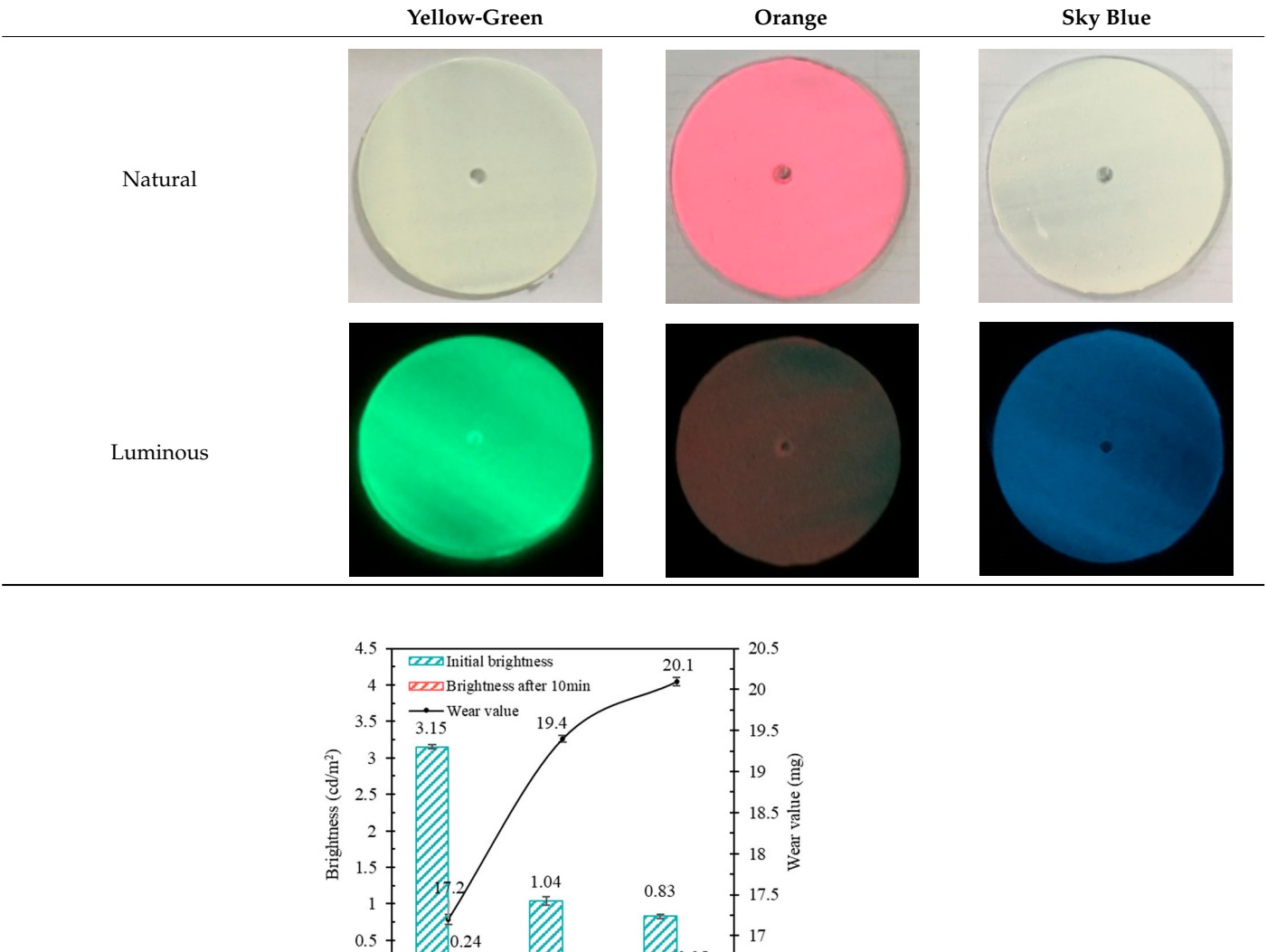

**Figure 3.** Effect of different ALP colors on coating performance.

The performance of coatings prepared with different ALP particle sizes was shown in Figure 4. As the particle size of ALP decreased, the performances of luminescence and afterglow became worse. This is why the large particle size resulted in large light absorption; the coating had better luminous performance and afterglow performance [47]. Moreover, the smaller the particle size, the better the adhesion and anti-abrasion properties. It was

concluded that the large particle size of ALP was not completely covered by the emulsion when preparing the coating, resulting in poor adhesion performance. Additionally, this made the surface of the coating too rough and increased the abrasion. For selecting the best particle size of ALP, the luminescence morphology of the coatings prepared with different particle sizes in the dark were compared, as shown in Figure 5. Overall, 100 mesh ALP-prepared coatings showed large black spots in the fluorescent state, while 300 mesh coatings gave uniform luminescence, because the luminescent powder with large particle size was prone to precipitation and poor dispersion after being prepared into coatings [48]. In contrast, the luminescent powder with the smaller particle size had better dispersion in the coating, and its surface was smoother after being prepared into the coating. Consequently, the 300-mesh yellow-green ALP was chosen as the optimal ALP.

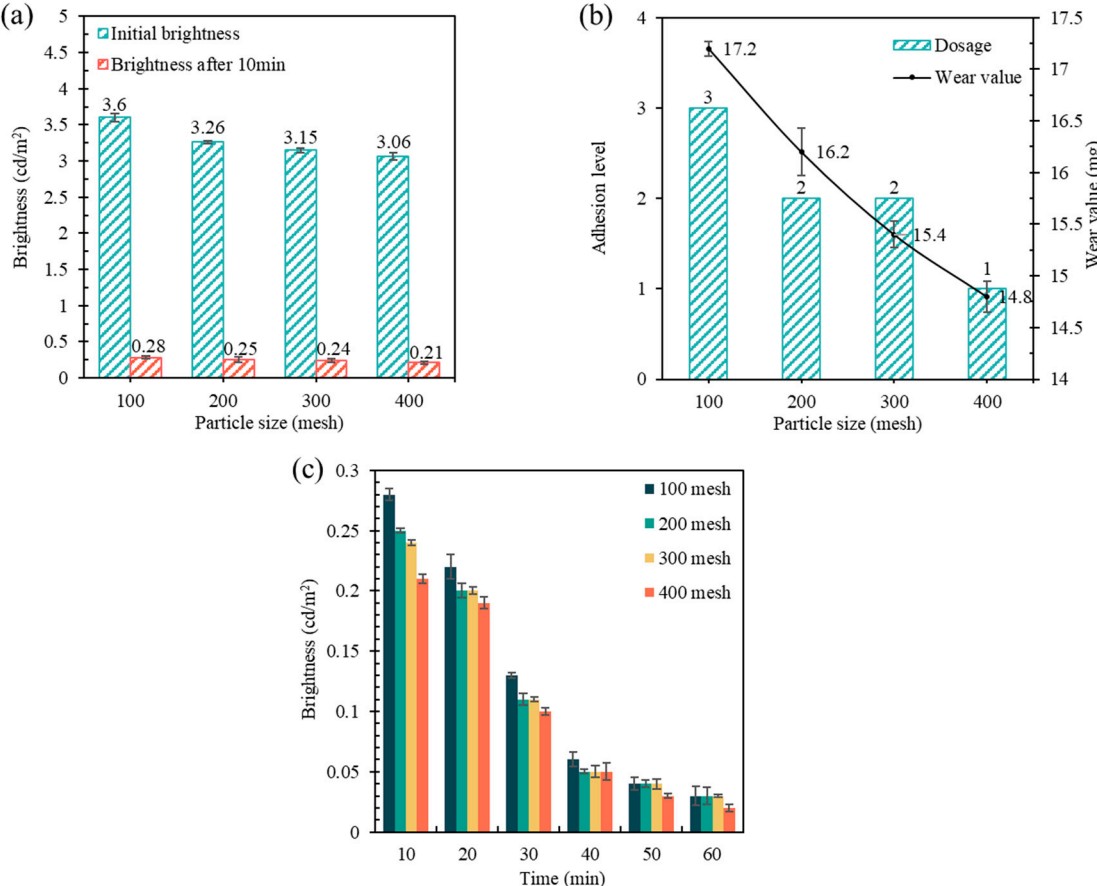

**Figure 4.** Effect of different ALP particle sizes on coating performance. ((**a**) luminous performance; (**b**) physical properties; (**c**) afterglow performance).

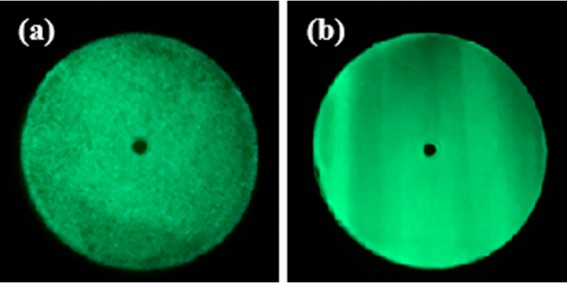

**Figure 5.** The morphology of coatings made of different particle sizes of luminescent powder ((**a**) 100 mesh; (**b**) 300 mesh).

### 3.1.2. Dosage of ALP

The components of the coatings included ALP, pigment, filler, and anti-sedimentation agent. The effects of different dosages of components on the performance of the coatings were compared based on the cost and technical specifications of road safety engineering, as shown in Figure 6. The initial brightness and afterglow of the luminous coating grew gradually with the increase in the luminescent powder dosage, but the growth rate decreased. Because ALP reaches a certain amount, the distribution of ALP in the thin coating film also tends to saturate, and the influence of its dosage on the luminous performance of the coating will gradually stabilize. Additionally, the increasing dosage reduced the wear value to some extent, but the influence value was not large. In addition, the adhesion was maintained at level 2 without change, since it was mainly determined by the contact area between the coating and the substrate. Therefore, the effect of luminescent powder dosage on the performance of luminous coatings was large. Considering the cost and abrasiveness, 25% was chosen as the best dosage.

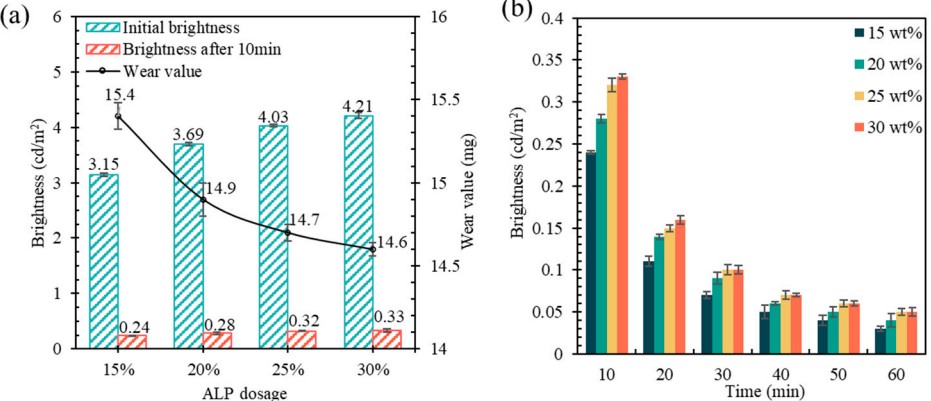

**Figure 6.** Effect of different ALP dosages on coating performance. ((**a**) luminous performance and physical properties; (**b**) afterglow performance).

### 3.1.3. Dosage of Film-Forming Auxiliaries

Film-forming auxiliaries mainly affect the emulsion and have almost no effect on the luminescent performance of the coating. Therefore, only their effects on the physical properties of the coating were considered, and the dosage of AE-12 used as a comparison test was determined as 2%, 3%, and 5% of the emulsion mass, and the test temperature was unified at 15 °C [41]. As shown in Figure 7, it can be seen that at 15 °C without AE-12 the coating could not form film, and the emulsion dried with cracking, peeling, yellowing, and other undesirable phenomena. When mixed with 2% of AE-12, the film-forming performance improved obviously, but there was still the phenomenon of local wrinkles. Furthermore, when the dosage of AE-12 increased to 3%, the appearance of the emulsion after film-forming was without any adverse characteristics.

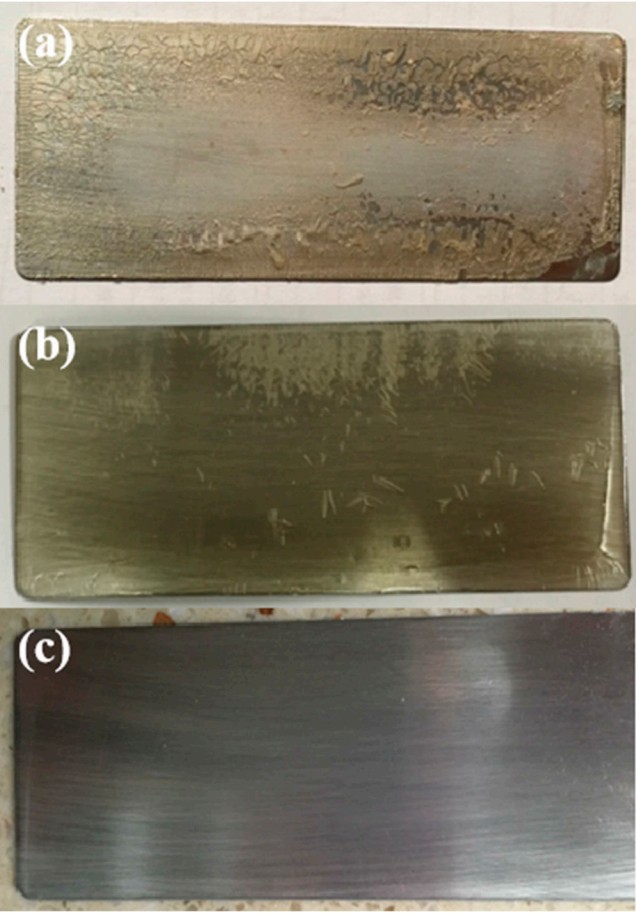

**Figure 7.** Appearance of paint specimens with different dosages of AE-12. ((**a**) 0 wt%; (**b**) 2 wt%; (**c**) 3 wt%).

Figure 8 presented the wear value decreases with the increase in AE-12 dosage, but the total abrasion was very small. This was mainly because other fillers, such as luminescent power and pigments, had not been added in the experiment, which form a uniform and dense paint film under the action of AE-12 and are hard to abrade. In terms of adhesion, the film-forming emulsion with 2% AE-12 had the largest wear value after film formation, which was mainly because the film-forming performance of the emulsion was not good at this time. Additionally, when 3% AE-12 was added, the adhesion no longer changed with the increase in the AE-12 dosage, which means that the adhesion, like the wear value, could also reach the ideal value under the premise of the excellent film-forming effect. It is worth mentioning that the average surface drying time of the emulsion was 25 min, and the longer drying time was mainly because the overall solid content was relatively low in the test sample without the addition of filler. After adding pigments and fillers in the later coating test, the solid content was significantly increased and the surface drying time was significantly shortened. After taking into account the wear value, adhesion level, surface drying time, and other performance effects, the optimal dosage of AE-12 was finally determined to be 3% of the SSAE.

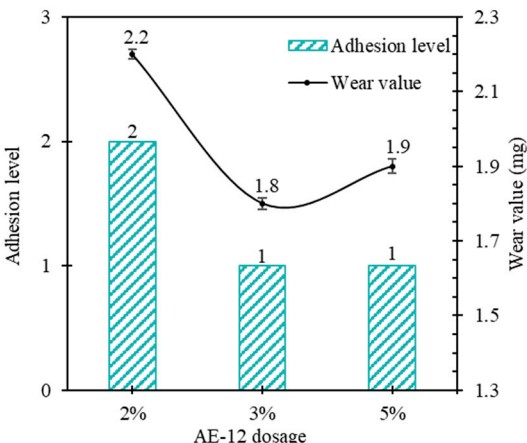

**Figure 8.** Effect of different AE-12 dosages on coating performance.

### 3.1.4. Dosage of Pigment

Figure 9a showed that the brightness and wear value of the prepared coatings increased with the increase in RTDP dosage, but the growth rate decreased gradually. Therefore, the long afterglow properties of the coatings prepared with different dosages of RTDP at 10 min, 20 min, 30 min, 40 min, 50 min, and 60 min, respectively, were compared in Figure 9b. The afterglow performance of the coatings blended with 7% RTDP gradually decreased with time, which may have been caused by the high covering property of titanium dioxide [49]. Images of the coating before and after the addition of 5% RTDP can be observed, as shown in Figure 10. The luminous marking coating without RTDP displayed a more transparent state, while the titanium dioxide as a white pigment provided a high masking effect. Furthermore, the abrasion value was reduced with the increase in RTDP dosage. From these, the RTDP dosage was determined to be 5%.

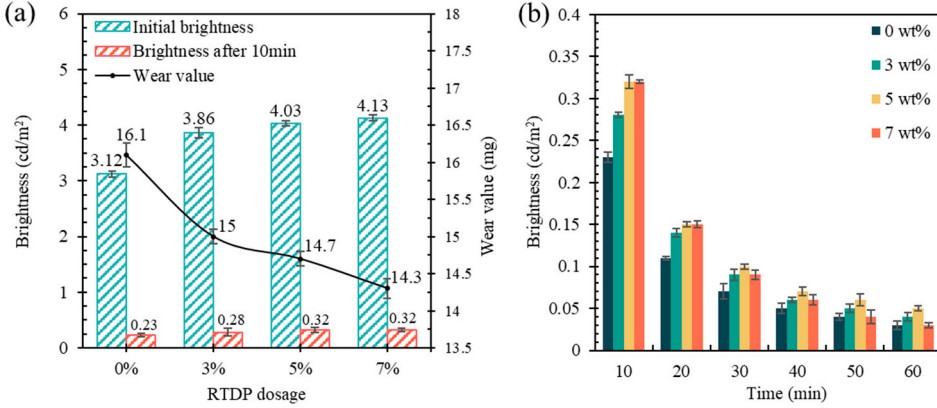

**Figure 9.** Effect of different RTDP dosages on coating performance. ((**a**) luminous performance and physical properties; (**b**) afterglow performance).

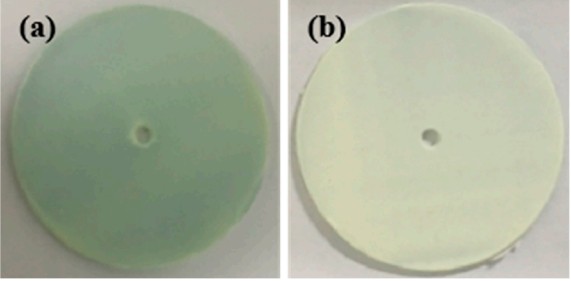

**Figure 10.** Appearance of coated specimens with different RTDP dosages. ((**a**) 0 wt%; (**b**) 5 wt%).

### 3.1.5. Dosage of the Filler

Since the UGP itself had good light transmission and reflective properties, the increase in its dosage would have a certain enhancement effect on the luminous powder's light absorption, light storage, and luminous effects. Figure 11 demonstrated that the dosage of UGP had a positive effect on the initial brightness and afterglow performance of the coatings. As the dosage of UGP increased from 40% to 50%, the dry film wear value of the luminous coating gradually decreased and the adhesion performance reached the 2 level. However, the coating abrasion value increased and the adhesion performance decreased when the UGP dosage reached 55%. This might be attributed to the filler amount having reached the saturation state, and the excessive addition having produced a negative effect on the coating performance. Therefore, the 50% dosage was selected as the best dosage of UGP in luminous marking coatings.

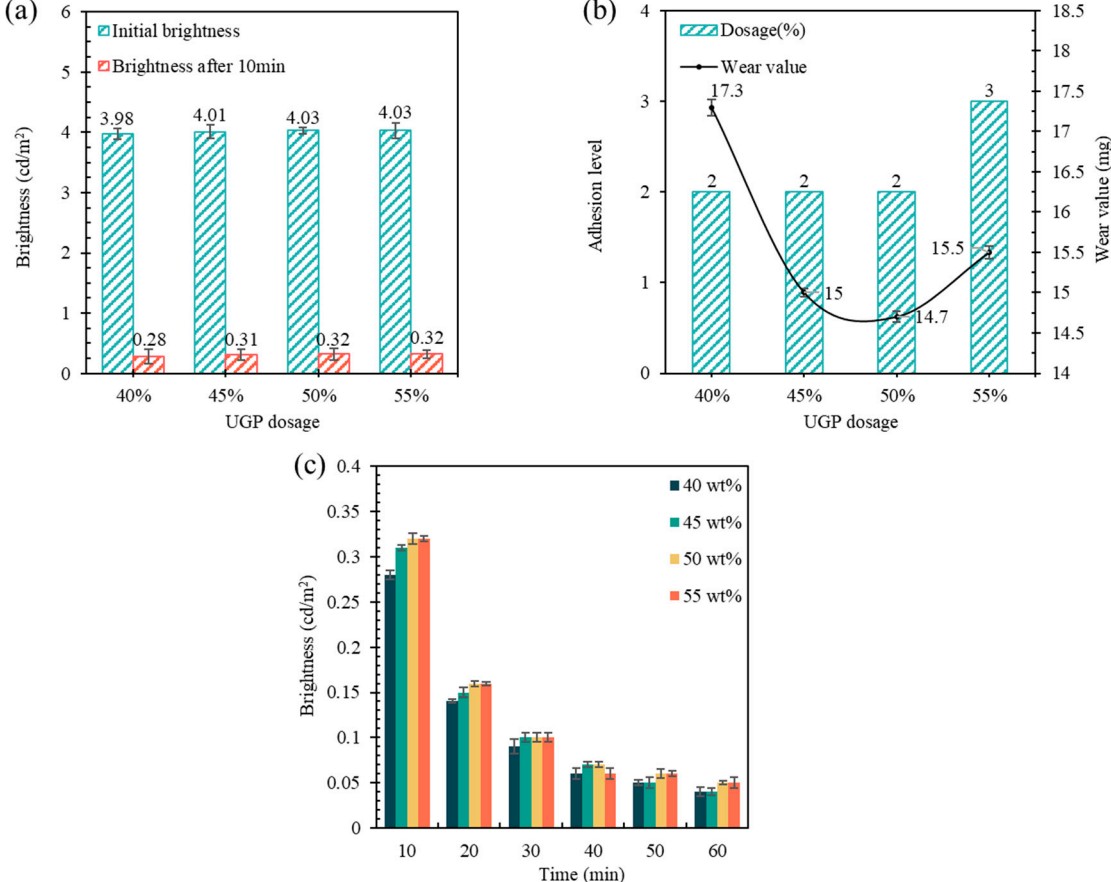

**Figure 11.** Effect of different UGP dosages on coating performance. ((**a**) luminous performance; (**b**) physical properties; (**c**) afterglow performance).

### 3.1.6. Dosage of FSP

The luminescence, anti-abrasion, and anti-sedimentation properties of the coatings with different dosages of anti-sedimentation agents are given in Figure 12. As the dosage of FSP increased from 0.6% to 0.8%, the initial brightness increased, and the afterglow performance became worse. For the observation of the effect of FSP dosage on the anti-sedimentation performance, the brightness of the paint after centrifugation for 15 min was compared in Figure 12b. After centrifugation for 15 min, the brightness of the coating increased with the dosage of FSP. The reason was that the dosage of anti-sedimentation agent in the coating system was too limited to form a sufficient three-perimeter mesh structure, which led to the weak dispersion of luminous powder. However, an excessive dosage of the anti-sedimentation agent was no longer beneficial to the luminescence

performance of the coating after 15 min of centrifugation. This was attributed to a large dosage of FSP forming too much mesh structure, which would lead to an increase in viscosity, which was also not conducive to the uniform dispersion of the luminous powder. It was indicated that too much anti-sedimentation agent would lead to an unstable light absorption and light storage performance of the coating. Overall, the 0.8% dosage was selected as the best dosage of FSP in the luminous marking coating.

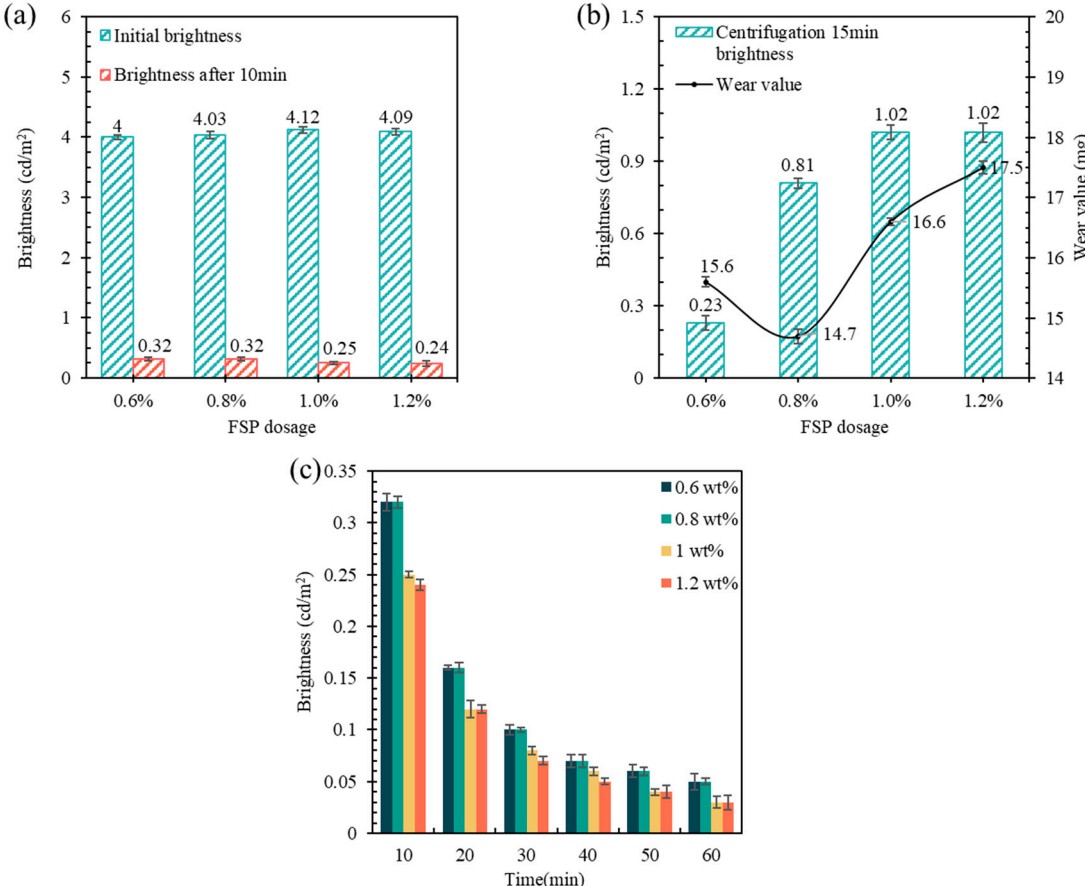

**Figure 12.** Effect of different FSP dosages on coating performance. ((**a**) luminous performance; (**b**) physical properties; (**c**) afterglow performance).

### 3.1.7. Dosages of MOP and FMP

As can be observed from Figure 13, in different dosages of defoamer and leveling agent, the initial brightness value of the paint was stable at about 4.02 cd/m$^2$ and the adhesion was stable at level 2, but the wear resistance of the paint was affected more. Both auxiliaries had the lowest wear values at 0.6% dosage. The loss of abrasion resistance was mainly caused by surface potholes. In Figure 14, when the dosage of MOP was 0.4%, the air bubbles generated during the mixing of paint could not be eliminated, resulting in small holes on the surface of the paint film after film formation. Additionally, when the dosage of MOP was 0.6%, there were no small holes on the surface of the paint film after film formation. The appearance of pores on the surface of the paint film was mainly related to the mechanism of action of the defoamer and leveling agent. On the one hand, due to the high solid content of the paint, bubbles were easily generated during the preparation process. However, when the dosage of MOP was 0.4%, the bubbles generated during the preparation process could not be eliminated due to the insufficient amount of MOP. After the paint film has been formed, some air bubbles on the surface will burst and form holes, while some others of the air bubbles present in the wet film will be retained and form bulges on the surface of the dry film. These holes and bulges are not conducive to the

abrasion resistance of the paint film. On the other hand, when the dosage of FMP was 0.4%, the leveling effect was limited and the flatness of the coating surface was not satisfactory, so the wear value of the coating with a 0.4% dosage was higher. Additionally, when the dosage was increased to 0.8%, the viscosity of the coating was found to increase to some extent during the preparation process, which may be caused by the poor compatibility of 0.8% FMP with other auxiliaries or emulsions.

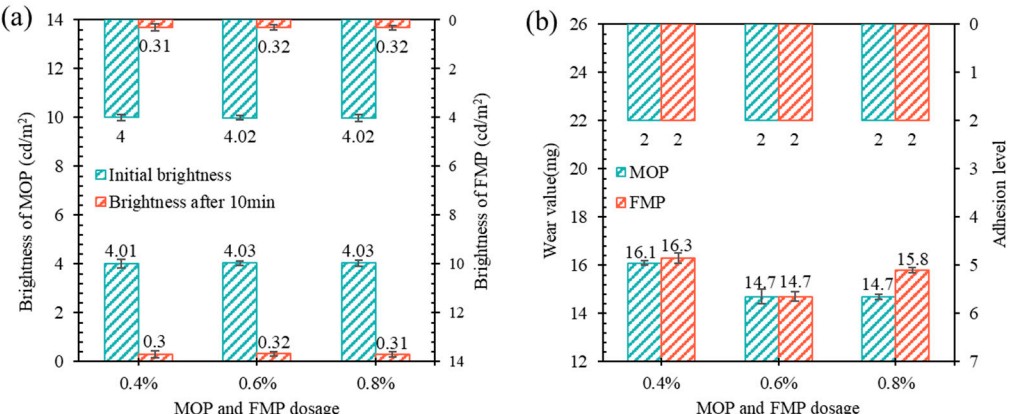

**Figure 13.** Effect of different dosages of MOP and FMP on coating abrasion performance. ((**a**) luminous performance; (**b**) physical properties).

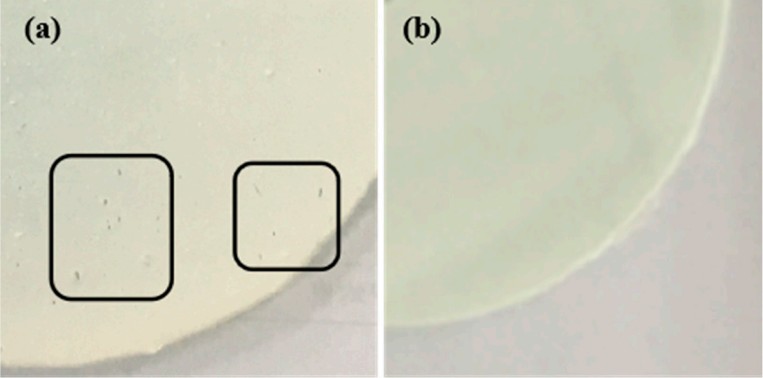

**Figure 14.** Appearance of paint film specimens with different MOP dosages. ((**a**) 0.4 wt%; (**b**) 0.6 wt%).

### 3.2. Multi-Factor Optimization Using RSM

The dosages of ALP, RTDP, and FSP were chosen as variables, and the brightness, brightness after 15 min centrifugation, and wear value were taken as response values to design 15 sets of experiments. The design table and experimental results are shown in Table 3. Data analysis requires the creation of a suitable model to avoid producing poor or incorrect results. Therefore, a quadratic model of responses was developed, and the analysis of variance (ANOVA) was used to determine F-values and $p$-values, as listed in Table 4. The ratio of the mean-squared sum of the factors in a row of the ANOVA table to the mean-squared sum of the errors is the F-value, and it describes the interaction significance and effect between the factors. The $p$-value is the quartile of the F-distribution of degrees of freedom, which corresponds to the F-value and is used for identifying factors in the model that have statistical significance. The significance level a = 0.05 was used in response surface analysis to investigate the significance of the factors. $p < 0.05$ indicates a significant correlation. Smaller $p$-values and higher F-values imply greater significance. It can be seen that the $p$-value of the $Z_1$-Brightness model was less than 0.05, indicating that the model was significant. The lack of fit was insignificant relative to the pure error, which implies that the model fit was successful. However, only the $p$-values of both A and $A^2$ parameters were less than 0.05, signifying that brightness was mainly influenced by ALP.

The model was re-optimized without considering other factors, and the fitting results were as Equation (3). Similarly, the $Z_2$-Brightness after centrifugation and $Z_3$-wear value models were redeveloped to fit the equations as in Equations (4) and (5).

$$Z_1 = 0.0076A^2 - 0.3295A + 7.5104 \tag{3}$$

$$Z_2 = 7.0179A^2 + 12.4411C + 0.0238A - 5.4538 \tag{4}$$

$$Z_3 = 0.0101A^2 + 0.2797B^2 - 0.1639AB + 0.4022AC + 1.4889B - 8.3056C + 11.1984 \tag{5}$$

**Table 3.** Experimental scheme and response values obtained from the Box–Behnken design and laboratory experiment.

| Run | Coded Variable | | | Real Variable | | | Response Value | | |
|---|---|---|---|---|---|---|---|---|---|
| | $X_1$ | $X_2$ | $X_3$ | A: ALP (%) | B: RTDP (%) | C: FSP (%) | $Z_1$ (cd/m$^2$) | $Z_2$ (cd/m$^2$) | $Z_3$ (mg) |
| 1 | 1 | 0 | −1 | 27 | 5 | 0.7 | 4.15 | 0.46 | 12.7 |
| 2 | 1 | −1 | 0 | 27 | 4 | 0.8 | 4.13 | 0.65 | 13.4 |
| 3 | 1 | 0 | 1 | 27 | 5 | 0.9 | 4.17 | 0.68 | 13.3 |
| 4 | 0 | 0 | 0 | 25 | 5 | 0.8 | 4.01 | 0.61 | 12.7 |
| 5 | 0 | 0 | 0 | 25 | 5 | 0.8 | 4.02 | 0.63 | 12.8 |
| 6 | 0 | −1 | 1 | 25 | 4 | 0.9 | 4.00 | 0.64 | 13.1 |
| 7 | −1 | −1 | 0 | 23 | 4 | 0.8 | 3.93 | 0.52 | 12.8 |
| 8 | −1 | 1 | 0 | 23 | 6 | 0.8 | 3.96 | 0.53 | 13.7 |
| 9 | −1 | 0 | −1 | 23 | 5 | 0.7 | 3.94 | 0.38 | 12.9 |
| 10 | 1 | 1 | 0 | 27 | 6 | 0.8 | 4.14 | 0.65 | 13.1 |
| 11 | 0 | 1 | −1 | 25 | 6 | 0.7 | 4.00 | 0.41 | 13.2 |
| 12 | 0 | 1 | 1 | 25 | 6 | 0.9 | 4.03 | 0.66 | 13.4 |
| 13 | 0 | −1 | −1 | 25 | 4 | 0.7 | 4.01 | 0.39 | 12.6 |
| 14 | −1 | 0 | 1 | 23 | 5 | 0.9 | 3.96 | 0.63 | 13.0 |
| 15 | 0 | 0 | 0 | 25 | 5 | 0.8 | 4.05 | 0.62 | 12.8 |

**Table 4.** ANOVA results of optimization models of responses.

| Source | $Z_1$ | | | $Z_2$ | | | $Z_3$ | | |
|---|---|---|---|---|---|---|---|---|---|
| | F-Value | *p*-Value | Sig. | F-Value | *p*-Value | Sig. | F-Value | *p*-Value | Sig. |
| Model | 40.8413 | 0.0004 | S. | 39.1348 | 0.0004 | S. | 40.4106 | 0.0004 | S. |
| A: ALP | 342.8571 | <0.0001 | S. | 40.5618 | 0.0014 | S. | 0.3261 | 0.5927 | Ins. |
| B: RTDP | 1.9286 | 0.2236 | Ins. | 0.7022 | 0.4402 | Ins. | 73.3696 | 0.0004 | S. |
| C: FSP | 1.9286 | 0.2236 | Ins. | 264.2978 | <0.0001 | S. | 63.9130 | 0.0005 | S. |
| AB | 0.4286 | 0.5416 | Ins. | 0.0562 | 0.8220 | Ins. | 93.9130 | 0.0002 | S. |
| AC | 0.0000 | 1.0000 | Ins. | 0.5056 | 0.5088 | Ins. | 16.3043 | 0.0099 | S. |
| BC | 1.7143 | 0.2474 | Ins. | 0.0000 | 1.0000 | Ins. | 5.8696 | 0.0599 | Ins. |
| A$^2$ | 13.4615 | 0.0145 | S. | 0.8297 | 0.4041 | Ins. | 35.3846 | 0.0019 | S. |
| B$^2$ | 3.9670 | 0.1030 | Ins. | 4.2005 | 0.0957 | Ins. | 81.9398 | 0.0003 | S. |
| C$^2$ | 0.0110 | 0.9206 | Ins. | 43.6128 | 0.0012 | S. | 0.2676 | 0.6270 | Ins. |
| Lack of Fit | 0.6060 | 0.7597 | Ins. | 6.7500 | 0.1318 | Ins. | 1.2500 | 0.4733 | Ins. |

Note: Sig. = significance; Ins = insignificant; S = significant.

The coefficients of determination for optimization models are provided in Table 5. $R^2$ values similar to 1 indicated that the regression model could fit the input factors and response values to a high degree. The high and near values of Adj.$R^2$ and Pred.$R^2$ ($\left| \text{Adj.} R^2 \text{Pred.} R^2 \right| < 0.2$) indicated that the regression model was suitable for analyzing the process. Adeq Precision measures the signal-to-noise ratio, and a ratio greater than 4 is

desirable. The Adeq Precisions of three models were greater than 4, indicating an adequate signal. These models could be used to navigate the design space.

**Table 5.** Coefficient of determination for responses.

| Coefficient of Determination | $Z_1$ | $Z_2$ | $Z_3$ |
|---|---|---|---|
| Std. Dev. | 0.0171 | 0.0741 | 0.0211 |
| Mean | 4.03 | 13.03 | 0.564 |
| C.V.% | 0.423 | 0.5686 | 3.75 |
| $R^2$ | 0.9598 | 0.9689 | 0.9691 |
| Adjusted $R^2$ | 0.9531 | 0.9456 | 0.9607 |
| Predicted $R^2$ | 0.9379 | 0.8946 | 0.9408 |
| Adeq Precision | 26.2129 | 21.2898 | 30.9324 |

Based on the above optimization model, a model graph of brightness, brightness after 15 min centrifugation, and wear value with ALP, RTDP, and FSP was established. Brightness was mainly influenced by ALP, and thus the model was a two-dimensional single-factor variation graph, as shown in Figure 15. The black square indicated the data point, and the solid black curve was the corresponding fitted line. In the range of 23%–27%, the increase in ALP dosage caused a gradual increase in brightness, and the degree of increase was also gradually increased, which differed from the conclusion of the univariate analysis. This may be because there were other factors as independent variables in the experimental design, thereby generating errors. However, the errors were not significant and did not affect the optimization results. The dosage of RTDP had no effect on the brightness after centrifugation. Because of the insignificance of RTDP, only the response surface model and contour graph under the two interactions of ALP-FSP were considered, as shown in Figure 16. The brightness after centrifugation increased with increasing ALP and FSP, and the response surface was steeper with FSP. This indicates that both had a positive effect on it, and the effect of FSP was more significant. There was no interaction between RTDP and FSP dosage on brightness after centrifugation, and the model graphs of wear value responding to ALP-RTDP and ALP-RTDP are shown in Figures 17 and 18. The surface of wear value with increasing FSP was the steepest, RTDP was the second, and ALP was the slowest. It can be concluded that wear value is influenced by ALP, RTDP, and FSP, and the degree of influence is RTDP > FSP > ALP. The ratio of the long and short axes of the ellipse in Figure 17b was larger than that of the ellipse in Figure 18b, which implied that the interaction between ALP and RTDP was larger than that between ALP and FSP, signifying that the wear value is most influenced by RTDP. The response surface plot allows the evaluation of the two-by-two interaction of the test factors on each response output to determine the optimal level range for each factor (the area near the response surface vertex). Multivariate optimization of ALP, RTDP, and FSP parameters can be used to obtain the maximum brightness, maximum brightness after centrifugation, and minimum wear value. It was found that the optimums of ALP, RTDP, and FSP were 27.000%, 5.193%, and 0.805%, respectively.

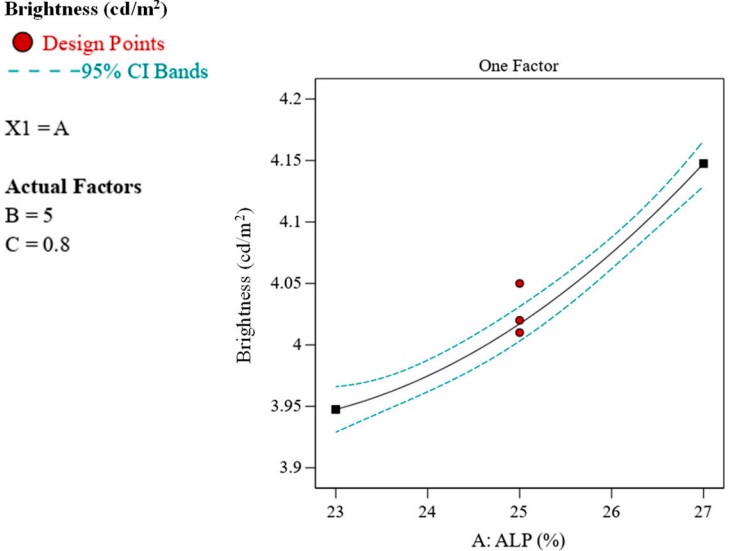

**Figure 15.** Single-factor graph of brightness to ALP.

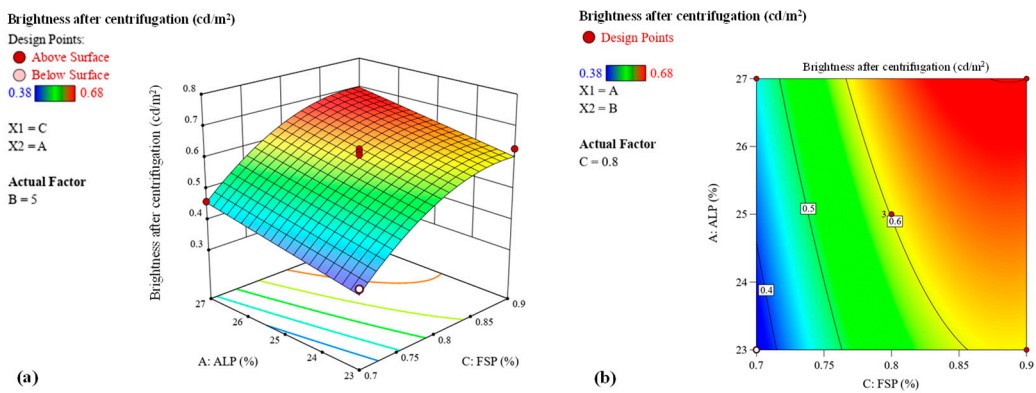

**Figure 16.** Response surface graph (**a**) and contour graph (**b**) of brightness after centrifugation to ALP and FSP.

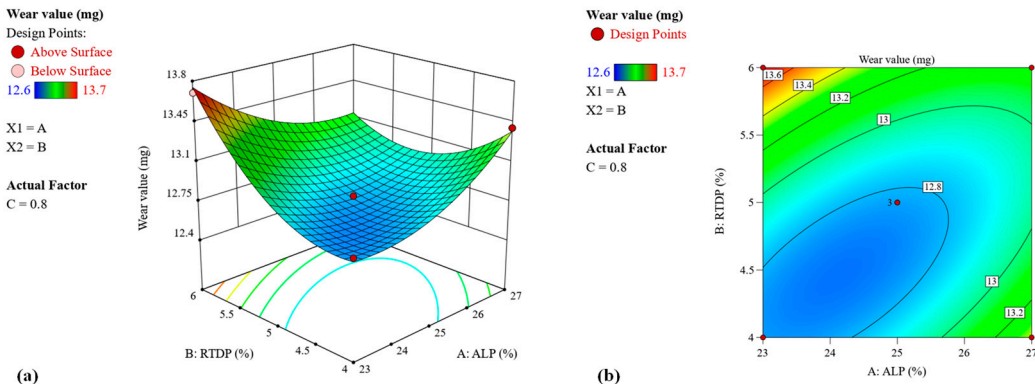

**Figure 17.** Response surface graph (**a**) and contour graph (**b**) of wear value to ALP and RTDP.

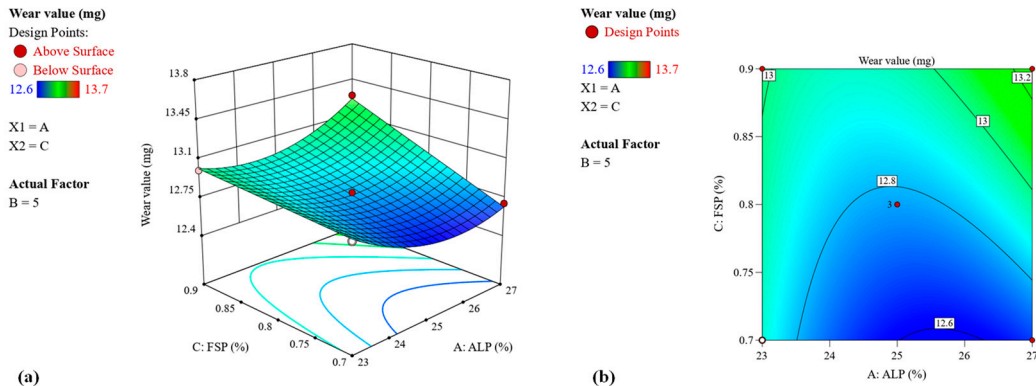

**Figure 18.** Response surface graph (**a**) and contour graph (**b**) of wear value to ALP and FSP.

### 3.3. Performance Characteristics of the Coatings

### 3.3.1. Physical Properties

The target coatings were prepared based on the optimal ratios preferred by the response surface method. The coating of traditional performance indicators following Road Marking Coating (JT/T 280-2004) was tested [50]. The results are reported in Table 6, and all meet the standard requirements.

**Table 6.** Physical properties of active luminous road marking coating.

| Test Items | Index | Experimental Results |
|---|---|---|
| State in container | No caking or crusting, and easy to stir | Qualified |
| Density | $\geq 1.4$ | 1.42 |
| Solid content | $\geq 70\%$ | 70% |
| Drying time | $\leq 15$ | 14.2 min |
| Coating appearance | There should be no wrinkles, spots, blistering, or cracking after the coating has been cured. | Smooth surface without blistering or other abnormal phenomena |
| Water resistance | No abnormality after soaking in water for 24 h | Nothing unusual |
| Alkali resistance | No abnormality after soaking in saturated limewater for 18 h | Nothing unusual |
| Abrasion resistance | 27% $\leq 40$ (JM-100 Rubber wheel) | 12.8 mg |
| Adhesive force | $\leq 5$ level | 2 level |
| Afterglow time | - | $\geq 8$ h |

### 3.3.2. Excitation and Emission Properties

Figure 19 displays the fluorescence excitation spectrum from 375 to 465 nm of the luminous marking paint coating, which contained the violet-blue part of visible light. It indicated that not only could UV light excite the prepared luminous marking coating, but the natural visible light also had a good excitation effect. The peak excitation wavelength was 420 nm, which belonged to the wavelength range of UV light; it indicated that UV light had the best excitation effect on the luminous marking coating. Natural light can excite the light-absorbing properties of the coating well, and the application in road safety engineering was universal, without providing special excitation conditions [25].

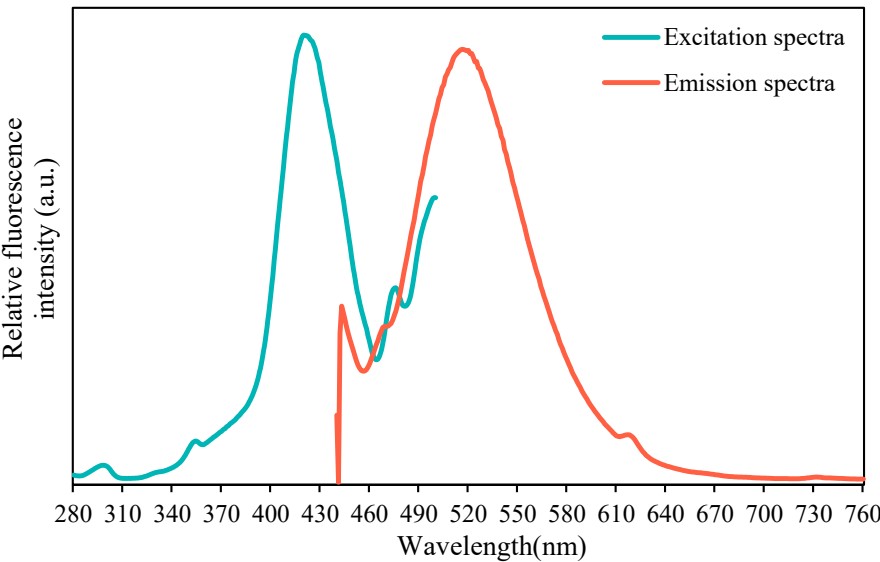

**Figure 19.** Fluorescence spectra of molecules within the coating in the excited and emitted states.

The spectral lines in the wavelength range of about 440 nm to 760 nm belong to the fluorescence emission spectrum. Four wave peaks appeared in the waveform diagram, corresponding to the wavelengths of about 443 nm, 470 nm, 516 nm, and 617 nm, of which the first spike was mainly caused by the irradiation of the light source and was not studied. The wavelength range of visible light was 400–760 nm; from the wavelength width, the fluorescence wavelength range of the luminous marking coating perfectly existed in the visible wavelength range, which means that its fluorescence can be well recognized by human eyes [51]. Additionally, the main peak was 516 nm, which belongs to the wavelength range of green light, the most sensitive color light for human eyes, indicating that the luminous marking coating prepared in this paper can play a high warning role in the dark. In summary, in terms of fluorescence spectrum, the luminous marking coating can play a helpful role in road safety engineering.

### 3.3.3. Micromorphology

Figure 20 shows the SEM images of the coating surface. A continuous coating structure had been formed on the surface of the coating paste, as seen in the scanned image on the 100 μm scale. Each raw material component was uniformly dispersed in the prepared coating. There were raised particles on the surface after the coating was formed, and these raised particles provided the surface roughness of the marking, increasing the tire–road friction, which was conducive to driving safety [3]. The scanned image from the 5-micron scale shows that the coating-former (SSAE) selected can wrap the powder granular material well and ensure the reliable durability of the luminous marking coating.

### 3.3.4. Optimization of Coating Thickness

Figure 21 shows the effect of coating thickness on the luminescence performance. The initial brightness of the luminous sign coating increased with the thickness of the coating; when the thickness reached 482 μm, the initial brightness reached a higher value of 4.17 cd/m$^2$. After that, the magnitude of the increase gradually decreased, and the initial brightness was 4.38 cd/m$^2$ when the thickness of the paint film was 546 μm, only 0.21 cd/m$^2$ higher than 482 μm. The main reason for the above phenomenon was that the phosphorescent material had been uniformly dispersed in the entire paint film, so the luminous body was not only on the surface part of the paint film, but also contained under the surface layer of the paint film. The presence of large dosages of RTDP and UGP in the coating system led to the coating not being completely light-transmittive [52,53]. As long as the coating thickness grew to a certain value, the luminous performance of the coating

was not completely excited by light, and the luminous performance also gradually reached saturation. The luminous marking coating in the practical application was determined to be about 500 μm.

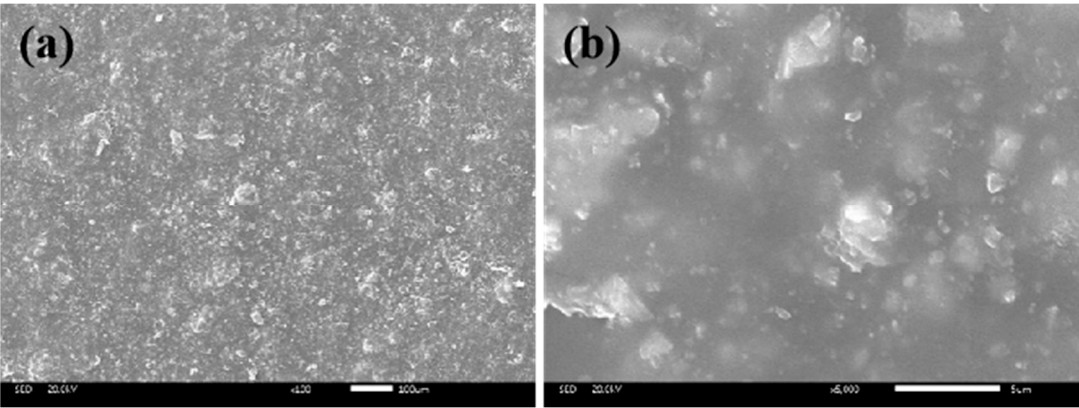

**Figure 20.** SEM images of luminous marking coating ((**a**) ×100; (**b**) ×5000).

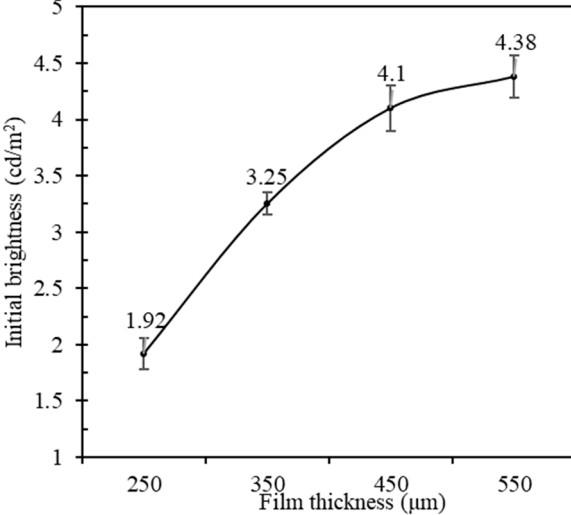

**Figure 21.** Initial brightness of different thicknesses of coatings under 10 min light.

### 3.3.5. Durability

Table 7 demonstrated the effect of the luminous marking coating under 1 h of rutting. It can be seen that after the rubber tires of the rutting instrument were crushed, black tire marks appeared on the surface of the marking coating, and the color of the marks was deeper after the high-temperature crushing. This was mainly because the asphalt was less viscous at high temperatures, which in turn produced cohesive damage on the surface of the mixture, resulting in more residual impurities such as asphalt on the tires [54]. In the whole test process, the luminous road marking coating rarely peeled, cracked, or displayed other undesirable phenomena. The stripping rate of the coating was only 8.2% under normal temperature rolling. Under high temperature rolling, the stripping rate of the coating increased to 16.8%, but it still had a relatively small value. This showed that the luminous road marking coating had good durability and would not easily produce undesirable symptoms such as shedding because of the vehicle crushing effect.

**Table 7.** The effect of 1 h rutting on the luminous marking coating.

| Temperature | Natural | Luminous | Stripping Rate |
|---|---|---|---|

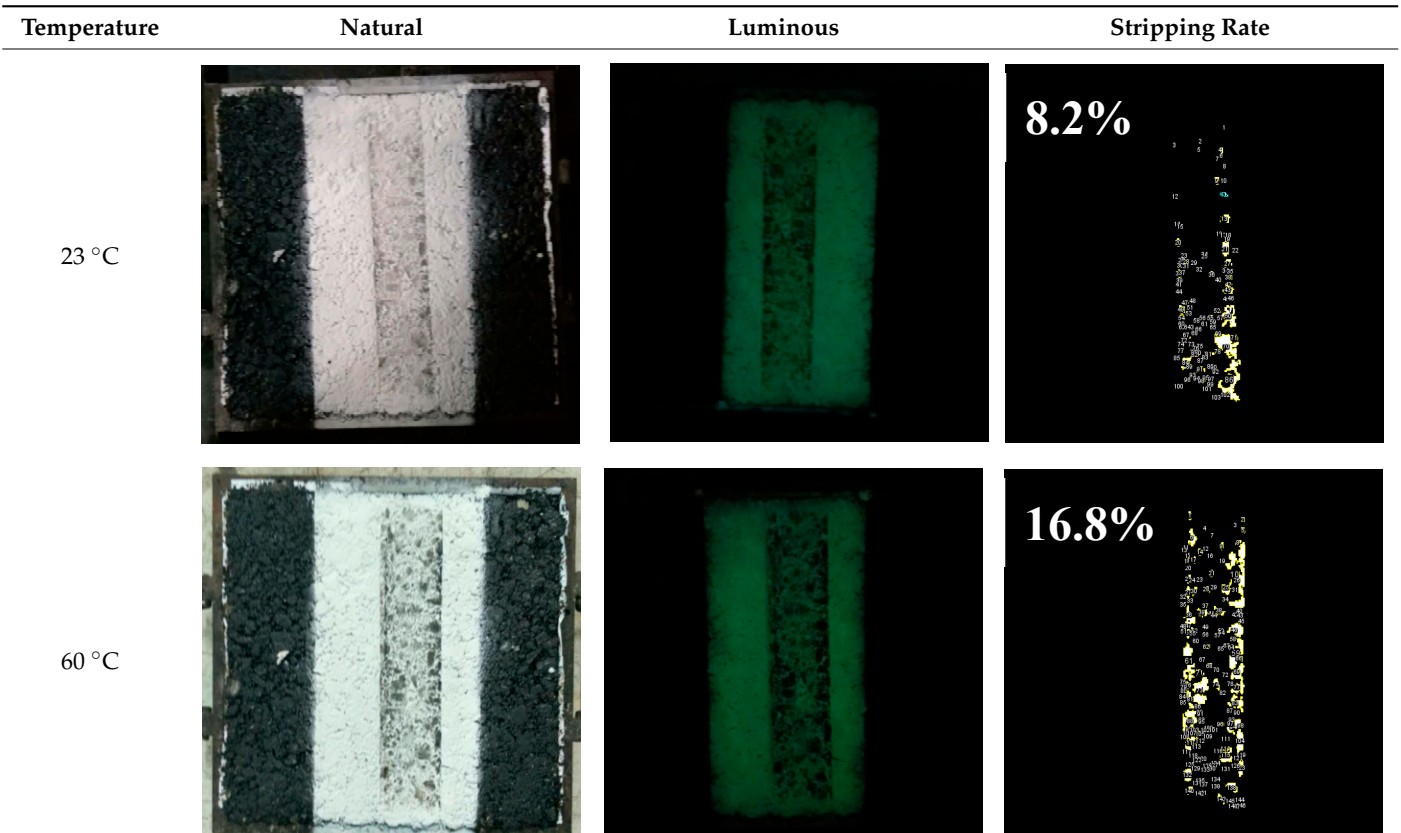

| | | | 8.2% (23 °C) |
| | | | 16.8% (60 °C) |

## 4. Conclusions

In this paper, a water-based luminous marking coating was prepared and the effect of the dosage of each component on the luminous, abrasion, and adhesion properties of the coating was investigated. The response surface method was adopted to obtain the optimal dosage of ALP, RTDP, and FSP. Coating performance, including basic performance, excitation and emission properties, micromorphology, coating optimum thickness, and durability, were characterized. The following conclusions can be drawn:

(1) Based on the corresponding single-factor analysis of each component in the coating, the color and particle size of the luminous powder were found to have different degrees of influence on the performance of the coating. For the coating prepared by using yellow-green luminous powder, the initial brightness reached 3.15 cd/m$^2$, which was more than three times the brightness of orange and sky-blue luminous powders. For comprehensive coating afterglow performance, adhesion performance and wear resistance, 300 mesh was considered the optimal particle size for luminous powder. Three factors had a greater impact on the luminescence, anti-sedimentation ability and wear resistance of luminous marking coatings: luminous powder dosage, fumed silica dosage, and titanium dioxide dosage.

(2) Combined with the corresponding single-factor analysis and response surface model analysis, it was found that brightness was mainly influenced by ALP, which increased with increasing dosage. ALP and FSP had a positive effect on the brightness after centrifugation, and the effect of the FSP dosage was more significant. The wear value was influenced by ALP, RTDP, and FSP, and the magnitude of the effect was RTDP > FSP > ALP. The interaction between ALP and RTDP was greater than that between ALP and FSP. The optimal dosages of each component were 27% ALP, 5% RTDP, 0.8% FSP, 50% UGP, 3% AE-12, 0.6% MOP, and 0.6% FMP.

(3) The MFS results demonstrated that the excitation spectrum of the prepared coating was in the range of 375–465 nm, with a peak wavelength of 420 nm, indicating that the

coating had the best excitation effect under UV light and was equally easy to excite under natural light. Its emission spectrum ranged from 440 to 760 nm, and the main emission peak, at 516 nm, indicated that the yellow-green light emitted from it could be sensitively recognized by human eyes. These imply that the luminous sign coating had a high warning effect in the dark and had great potential for applications in road safety engineering.

(4) The results of SEM showed that the components in the coating were evenly dispersed, and the rough surface provided a certain anti-slip property. When the coating thickness was increased from 482 μm to 546 μm, the initial brightness gradually reached 4.38 cd/m$^2$, and the optimal application thickness of the luminescent coating was determined to be 500 μm. After rutting tests at high or normal temperatures, essentially no undesirable phenomena such as stripping or cracking occurred. The stripping rates of the luminous marking coating were 16.8% and 8.2%, respectively, which indicated its excellent durability.

**Author Contributions:** K.W.: Conceptualization, Project administration, Writing—review and editing. Z.L.: Data Curation, Methodology, Writing—original draft. Y.Z. (Yingxue Zou): Software, Visualization. Y.Z. (Yunsheng Zhu): Supervision, Validation, Formal analysis. J.Y.: Investigation, Resources. All authors have read and agreed to the published version of the manuscript.

**Funding:** This research was funded by the National Natural Science Foundation of China [52178437].

**Institutional Review Board Statement:** Not applicable.

**Informed Consent Statement:** Not applicable.

**Data Availability Statement:** The data presented in this study are available on request from the corresponding author.

**Acknowledgments:** This research was supported by the National Natural Science Foundation of China [No. 52178437].

**Conflicts of Interest:** The authors declare no conflict of interest.

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
