# Peer review of "Preparation and Performance Characterization of an Active Luminous Coating for Asphalt Pavement Marking"

_coatings, doi:10.3390/coatings13061108_

Round 1
Reviewer 1 Report
A few more latest articles can be added to the research gap
English grammar needs improvement.
English grammar needs improvement.
Author Response
Thank you for your review suggestions. We have carefully revised the article, details of which are attached.

Reviewer 2 Report
The authors have performed an in-depth laboratory research to develop and evaluate a phosphorescent marking coating.
1. The title of the manuscript is misleading.
The title of the manuscript as it stands now is:"Optimization and Characterization of an Active Luminous Coating for Asphalt Pavement Marking". The researcher have not evaluated the performance of the coating on actual asphalt pavement subjected to actual traffic under different environmental conditions and at different times of the day. Accordingly, the title needs to be changed to provide a more realistic description of the contents of the manuscript.
2. Lines 30 - 32, 121 - 124, and 580 - 582.
These sentence cannot be supported by the findings of the research since no field performance tests were conducted. Their statements and conclusions to the results of the laboratory experiments without making bold and unsubstantiated statement and conclusions.
3. Line 32.
The research work is not "THEORETICAL". It is an "EXPERIMENTAL" work.
4. Lines 106 - 111.
I quote from the manuscript:
"Although the former people have some progress in the luminous properties of the accumulating luminous materials themselves, the effect of its play in the paint is not deep enough to study the impact [26]. At the same time, in the preparation process of luminous marking coatings, there are few researches on the compatibility of luminescent materials, pigments, fillers, and other auxiliaries."
I have several comments regarding this paragraph.
a. It is written in poor English.
b. What is meant by ".....not deep enough....."? The researchers need to assess the level of the depth of the specific research work done by others, and why the results of these works are not adequate.
c. This comment is a continuation of comment b above. The authors need to present justification for performing the proposed work and specifying its novelty and the research gap their work addresses.
5. The use of the word "AUXILIARIES".
The manuscript uses the word "AUXILIARIES" without discussing specifics (lines 102, 111, 116). Additionally, in line 116, it is mentioned that four kinds of auxiliaries are used in the research without naming them. Also, in Section 2.1.4, only two kinds of auxiliaries are mentioned.
6. Reorganization of the manuscript
I recommend organizing the manuscript according to the following outline:
* Abstract
* Introduction: The introduction usually does not contain literature review. It simply describes the background information.
* Objective: This section clearly states the objective of the research in a very concise way. The objective cannot be just reporting results of experiments.
* Literature Review: The reviewed literature needs to be relevant to the stated objective above. It is not enough just to enumerate the references. Each reference needs to be critically reviewed for its contributions and shortcomings
* Novelty and Uniqueness: Based on the literature review, this section provides a justification for conducting the research in the manuscript. The justification is based on its novelty and uniqueness. The research cannot be a duplication of what has been done in the past by other researchers.
* Methodology: This section describes how the objective(s) of the research will be achieved. Specifically, a list of tasks is identified for the execution of the research.
* Materials, Test Procedures and Specimen Preparation
* Results and , Analysis of Results: This section presents the results of the analyses.
* Conclusions and Recommendations
========================================================
Need extensive editing.
Author Response
Thank you for your review suggestions, we have carefully revised the article, details of which are attached.

Reviewer 3 Report
The current research work presents useful assessment and considerable contribution to the proposed field. Hence, manuscript requires mandatory revisions based on the following aspects:
1. Elaborate the novelty statement in detail.
2. More interpretation on results obtained can be written in the discussion section.
3. The discussion does not adequately relate this research to past reported research as well as to the relevant theoretical models. The paper is significantly lacking in scientific explanations for the observations reported. A deeper discussion of results is completely missing in the manuscript.
4. The present work in perspective of the currently available literature and the major breakthrough of this study in relation to the existing knowledge has not been properly discussed in the manuscript. The entire point is to look at the major findings of the current study in relation to the existing knowledge, however qualitative/quantitative.
5. The conclusions are not logically related to the data and the data are not sufficient to support the conclusions. In other words, the conclusions are stated as simple declarative sentences.
Moderate Editing of English Language is required.
Author Response

(The authors gave the same response as above.)

Reviewer 4 Report
1. This paper discusses the feasibility studies on the feasibility of Active Luminous Coating for Asphalt Pavement Marking.
2. In the abstract it is mentioned that the luminous road marking coating has good durability and will not produce undesirable symptoms such as peeling caused by vehicle crushing. The research findings need to be presented in quantitative terms rather than in qualitative terms.
3. In the Introduction section, proper research gaps need to be identified. Accordingly, objectives of the study are to be stated. Further novelty of the work need to be elaborated.
4. In the section 2.2.3, ASTM standards on brightness test, paint film abrasion test, and coating adhesion test need to be presented.
5. Authors need to justify on why the dosage of ALP, RTDP pigment, UGP and FSP filler were restricted to 30%, 7% and 55% respectively?
6. How many samples were tested for each composition to check repeatability of the results obtained?? For the experimental works carried out on different tests uncertainty error analysis must be presented. Error bars need to be added in the graphs.
7. Results and discussion section needs co-relation for the results presented to the already published literature.
8. Conclusions section can highlight on the important findings of the work reported.
9. The manuscript needs careful corrections for the English grammar and typo errors.

Moderate editing of English language required.
Author Response

(The authors gave the same response as above.)

Round 2
Reviewer 2 Report
The authors have responded satisfactorily to my comments. Accept as is.
Reviewer 3 Report
Authors modified the manuscript by incorporating most of the reviewer's comments and hence manuscript can be accepted for publication.
Minor Editing of English language is required.